# A Modular Genetic Approach to Newborn Screening from Spinal Muscular Atrophy to Sickle Cell Disease—Results from Six Years of Genetic Newborn Screening

**DOI:** 10.3390/genes15111467

**Published:** 2024-11-13

**Authors:** Jessica Bzdok, Ludwig Czibere, Siegfried Burggraf, Natalie Pauly, Esther M. Maier, Wulf Röschinger, Marc Becker, Jürgen Durner

**Affiliations:** 1Department of Operative/Restorative Dentistry, Periodontology and Pedodontics, Ludwig-Maximilians-Universität München, 80336 Munich, Germany; 2Laboratory Labor Becker MVZ eGbR, 81671 Munich, Germany; 3TIB Molbiol Syntheselabor GmbH, 12103 Berlin, Germany

**Keywords:** spinal muscular atrophy (SMA), sickle cell disease (SCD), severe combined immunodeficiency (SCID), newborn screening (NBS), cystinosis, qPCR, T-cell receptor excision circles (TREC), prevalence

## Abstract

Background/Objectives: Genetic newborn screening (NBS) has already entered the phase of common practice in many countries. In Germany, spinal muscular atrophy (SMA), severe combined immunodeficiency (SCID) and sickle cell disease (SCD) are currently a mandatory part of NBS. Here, we describe the experience of six years of genetic NBS including the prevalence of those three diseases in Germany. Methods: Samples and nucleic acids were extracted from dried blood spot cards, commonly used for NBS. A qPCR assay was used to detect disease-causing variants for SMA and SCD, and the detection of T-cell receptor excision circles (TRECs) was performed for SCID screening. Results: The results of the NBS of over 1 million newborns for SMA, approximately 770,000 for SCID and over 410,000 for SCD are discussed in detail. In these newborns, we have identified 121 cases of SMA, 15 cases of SCID and syndrome-based immunodeficiencies and 77 cases of SCD or β-thalassemia. Conclusions: The flexibility of multiplex qPCR is assessed as an effective tool for incorporating different molecular genetic markers for screening. The processing of dried blood spot (DBS) filter cards for molecular genetic assays and the assays are described in detail; turn-around times and cost estimations are included to give an insight into the processes and discuss further options for optimization. The identified cases are in the range expected for the total number of screened newborns, but present a more exact view on the actual prevalences for Germany.

## 1. Introduction

Newborn screening (NBS) has evolved rapidly over the past decades and recently included conditions that are only detectable by molecular genetic approaches. For all screened conditions, there is a growing need to develop and optimize techniques that are easy and reliable to perform at an acceptable cost. Although screening programs vary from country to country, there are some commonly accepted principles that need to be fulfilled [1,2,3]. One of the key principles is an existing treatment option for the disease screened. Blood collected by heel pick during the first days of life (in Germany, between 36 and 72 h after birth) and dried on special blood-collecting filter paper is sufficient to diagnose the most commonly screened congenital disorders. With growing knowledge of the pathogenesis and novel treatment options available, a number of new diseases have been considered or added to NBS programs [4,5,6,7].

Since some of those conditions can only be detected by analyzing the respective genes or DNA-containing by-products, molecular genetic analysis has become another pillar of NBS, in addition to the measurement of various metabolites, enzyme activities or hormones. Genetic testing in the official German NBS program currently includes spinal muscular atrophy (SMA) and severe combined immunodeficiency (SCID). Previously, we also had cystinosis included in a pilot project for genetic NBS [5]. For sickle cell disease (SCD) screening, high-performance liquid chromatography (HPLC) or matrix-assisted laser desorption/ionization (MALDI) is commonly used. However, a combination of genetic testing with confirmation by HPLC appears to be more cost- and time-efficient for high-throughput screening [8].

Infants with SCID are highly susceptible to severe infections, with an estimated mortality rate of nearly 100% within the first two years of life in the absence of treatment. SCID is the collective term used to describe a group of known rare genetic disorders that lead to the absence or very low numbers of T-cells. T-cell receptor excision circles (TRECs) are episomal DNA by-products that arise from the recombination of T-cell receptor DNA during thymopoiesis. Therefore, low or absent TRECs are indicative of SCID and other forms of T-cell lymphopenia (TCL). Neonatal TREC levels determined by TREC-specific PCRs can be used to detect impaired T-cell development and hence to screen for SCID [7,9,10,11].

SMA is an autosomal recessive neuromuscular disease with a high morbidity and mortality rate without treatment [12]. The prognosis for newborns affected by this disease has undergone a significant transformation with the introduction of therapeutic options such as Zolgensma^®^ or Spinraza^®^/Nusinersen [13,14]. The survival of motor neuron (*SMN*) genes is encoded on chromosome 5, with two highly homologous copies called *SMN1* and *SMN2* that differ by five bases that result in an alternative splicing event. The transcription of *SMN1* results in the generation of full-length mRNA, whereas the majority of *SMN2* transcripts lack exon 7 due to alternative splicing. A homozygous deletion of exon 7 or both exon 7 and exon 8 in the *SMN1* gene is known to be the main genetic cause of SMA with a prevalence of >95%. It has been observed that small amounts of *SMN2* transcripts are spliced into full-length transcripts. Thus, higher allelic copy numbers of *SMN2* usually delay the onset of the clinical signs and symptoms of SMA [15].

The term SCD encompasses a group of hemoglobin disorders that are inherited in an autosomal recessive manner. These disorders are characterized by the presence of hemoglobin S (HbS), which is translated due to a point mutation in the sixth codon of the hemoglobin subunit β (*HBB*) gene causing a Glu > Val substitution in HBB [16]. The underlying genotype of the disorder can be either homozygous or compound heterozygous, depending on the presence of a second pathogenic HBB mutation. In affected newborns, the variant HbS, rather than the physiologic hemoglobin (HbA), increases in the first few months of life. Once a critical amount of HbS is reached, typically between the third and fourth month of life, symptoms, e.g., anemia and pain due to vaso-occlusive events, begin to manifest in individuals affected by SCD [17].

Cystinosis is a lysosomal storage disorder leading to the loss of renal function within the first decade of life. It is autosomal recessive with the most disease-causing variants encoded in the cystinosin (*CTNS*) gene. Screening for the most common variants can help to identify most of the patients, who can then be treated by administering cysteamine [5,18].

Dried blood spots contain enough DNA to be used for many different genetic assays to be screened by NBS. Multiplex modular genetic NBS assays keep the costs, time and equipment required for screening at an affordable level [4,5], and provide the flexibility to change or replace targets. However, a detailed test of the assay performance with these changes always requires a careful assessment.

Here, we present the detailed results of six years of genetic newborn screening, the achieved turn-around times and a guideline for the implementation of these and other novel genetic tests for high-throughput genetic newborn screening.

## 2. Methods

### 2.1. Samples and Preparation of DBSs

The first pilot study for SMA and cystinosis was approved by the local ethics commission (Ethics Commission No. 16125 of the Bayerische Landesärztekammer, Munich, Germany) for the screening of SMA and cystinosis. The samples were mainly recruited from hospitals, with approximately 42% of the samples originating from the State of North Rhine-Westphalia (northwest Germany) and 58% from Bavaria (south Germany) [4]. Samples for routine testing after the pilot study phase were acquired from the same hospitals, and parents were informed and asked to agree or disagree with participation in the NBS program. The 3.2 mm punches were acquired from DBS cards (TFN, Ahlstrom-Munksjö, Bärenstein, Germany) using the Panthera Puncher 9 (Perkin Elmer, Waltham, MA, USA) and distributed to 96-well plates. To rule out the risk of a plate mix-up or high contamination, no-template controls were used, alternating between various positions on each plate. Special adapters were required to fit 96-well semi-skirted PCR plates (4ti-0770/C, 4titude, Surrey, UK, or 04-083-0150, Nerbe plus, Winsen, Germany). The plates were sealed using a qPCR seal (4ti-0560, 4titude, or Azenta Life Sciences, Griesheim, Germany) for transport between punching and nucleic acid isolation.

### 2.2. Rapid Nucleic Acid Extraction from Dried Blood Spots (DBSs)

The nucleic acid extraction from DBSs was performed as published in [4] and modified by [19] using a ViaFlo96 system (Integra Biosciences, Zizers, Switzerland). Briefly, DBS punches were incubated twice in 96-well plates with 50 µL of water, then with 150 µL of CX buffer (1× PBS and 0.5% Thesit^®^) for 10 min each on a plate shaker (IKA Labortechnik, Staufen, Germany, or Heidolph Rotamax 120, Heidolph, Schwabach, Germany). Plates were then centrifuged at 1000 rpm for 2 min (Rotanta 460, Hettich, Tuttlingen, Germany) and the supernatant removed using an eight-channel vacuum device (Vacusafe, Integra, Biebertal, Germany). A washing step with 150 µL of water followed, then the water was removed after centrifugation and 50 µL of CE buffer (10 mM Tris, 0.25 mM EDTA and 2 mM NaOH; pH 11) was added to each well. The pH value of the buffer was roughly adjusted using pH indicator stripes (Merck, Darmstadt, Germany). The plates were then sealed using PCR sealing foil (Nerbe Plus) and frozen for a minimum of 20 min at −70 °C. To thaw the solution and release the DNA, the plates were placed in a thermal cycler (Applied Biosystems 2720, Foster City, CA, USA) for 10 min at a temperature of 92 °C.

### 2.3. qPCR

The dried LightMix^®^ TREC SMA HBB Newborn Assay (40-0621-44, TIB Molbiol, Berlin, Germany) was prepared according to the manufacturer’s instructions. The reaction mix consisted of 0.5 µL of LightMix^®^ TREC SMA HBB Newborn Assay, 2.0 µL of Multiplex DNA Master (07339577001, Roche, Mannheim, Germany) and 4.5 µL of H_2_O, giving a total of 7 µL per well, and was transferred to 384-well plates (04729749001, Roche). A total of 3 µL of DNA eluate was added with the ViaFlo96 system (Integra). qPCRs were performed using LightCycler^®^ 480 II instruments (05015243001, Roche) with the run profile as follows: Initial denaturation was performed at 95 °C for 10 min, followed by amplification for 45 cycles in three steps at 95 °C for 5 s, 61 °C for 10 s and 72 °C for 15 s. Fluorescence signals were measured once per cycle in the 61 °C step. A final melting step (95 °C for 30 s; 45 °C for 2 min; 75 °C with the continuous measurement of fluorescence and a ramping rate of 0.19 °C/s) was included in the protocol for the detection and genotyping of HBB.

For the analysis of the qPCR fluorescence signals, the fit point (for the SCID assay) or second derivative maximum analysis (for the SMA assay) of the LightCycler^®^ software (Roche) was applied.

The limit of detection (LoD) was defined for TRECs and *SMN1* by testing standard rows based on plasmid DNA with copy numbers from 0.01 to 1000 per reaction in eightfold replication. A calibration curve was generated from which the minimum detectable concentration was determined. For *HBB*, a dilution series of placental DNA (9007-49-2, Sigma-Aldrich, Taufkirchen, Germany) was analyzed.

In order to determine the linearity of the individual tests, the standard series was extended to 1,000,000 copy numbers per reaction. The calibration curve was generated and the linear range of the expected concentrations of the standard series was evaluated. The value of the coefficient of determination (R^2^) was calculated using LightCycler^®^ 480 Software (R^2^ = 1 − error). The efficiency resulted from the slope of the regression line and was calculated using the following equation.
$$\mathrm{efficiency}=(10^{\left(\frac{-1}{\mathrm{slope}}\right)}-1)\times 100\%$$

### 2.4. Assay Characteristics

The LightMix^®^ TREC SMA HBB Newborn detects the wild-type variant of *SMN1* c.840C>T (rs1164325688, ClinVar identifier: NM_000344.4 (SMN1):c.840C>T (p.Phe280=)). If this reaction shows an amplification signal, at least one functional copy of *SMN1* should be present or be expressed, thus 5q-associated SMA can be excluded. Heterozygous carriers of an *SMN1* deletion cannot be detected using this assay. In the case of *SMN1* testing, control DNA containing only *SMN2* copies is a useful control parameter as *SMN1* and *SMN2* differ by only five bases. Therefore, the false amplification of *SMN2* by the *SMN1* assay can be ruled out. Comparable to *SMN1*, the TREC analysis detects target sequences present only in wild-type samples, with signal loss in samples without TREC production. Thus, the *SMN1* and TREC assays served as internal controls for one another. The TREC assay was based on the detection of signal joint TRECs as described by Cossu [11]. Samples without TREC amplification (flatline) with two repeated measurements from a fresh DBS punch showing the same characteristics were classified as urgent positives for SCID. All other samples with late amplification signals with Cq-values > 31 as described previously [19] were considered for retesting, requesting a fresh DBS (control) card (recall) from the institution of origin. If retesting of a recall DBS card showed no or a late amplification signal, the sample was also considered urgently SCID-positive. Establishing laboratory-specific cut-offs and retesting is of special importance for SCID testing, as dust particles generated during the DBS punching process are a source of contamination and this risk must be assessed during validation [19,20]. To ensure consistency from batch to batch, the aliquot of a defined sample (positive control with plasmid DNA) was included in each run; Cq-values had to be within a range of ±0.5log_10_.

*HBB* is detected and genotyped by using a melting curve analysis. The wild-type-specific probe covers codons 3 to 11, which carry ~90% of all globin variants. This includes the more frequent variant rs334 (T>A), ClinVar identifier: NM_000518.5(HBB):c.20A>T (p.Glu7Val), encoding the HbS allele and the less frequent variant rs33930165 (C>T), ClinVar identifier: NM_000518.4(HBB):c.19G>A (p.Glu7Lys), encoding the HbC allele. If the wild-type-specific melting peak is missing, both alleles are either mutated or deleted. A heterozygous sample, in contrast, is characterized by a double peak. Samples exhibiting non-wild-type peaks or the absence of a peak were considered to be potentially pathogenic. Using heterozygote control samples helped to discriminate between wild-type and non-wild-type samples. In order to further characterize potentially pathogenic variants, the respective samples were screened in a second-tier test using automated high-performance liquid chromatography (HPLC) systems (HLC-723G8, TOSOH, Stuttgart, Germany) according to the manufacturers’ instructions.

### 2.5. Ethics Statement

The retrospective study of anonymously analyzed data was conducted in accordance with the Declaration of Helsinki. The study protocol was presented to the Ethics Committee of Ludwig Maximilians University Munich, Germany. The committee issued a letter of no counseling obligation for the study presented here (reference: 24-0351 KB).

### 2.6. Data Collection

Data were collected anonymously between 22 May 2024 and 6 June 2024, utilizing a statistics tool (Quickstat, medat computersysteme, Munich, Germany) for the retrospective study. Quantitative data were generated using the LightCycler^®^ software (Version 1.5.1.62 SP3). The data were generated between 7 June 2021 and 27 August 2021, 3 April 2023 and 28 April 2023, and 28 May 2024 and 30 June 2024, respectively.

## 3. Results

In 2018, we launched a large-scale pilot genetic screen for SMA and cystinosis in our laboratory [4,5,21] (Figure 1). After about two years of screening comprising about 200,000 samples, the screening for cystinosis was discontinued due to the low prevalence of about 1:100,000 [22]. At that time, screening for SCID became part of the NBS program in Germany. Therefore, within the multiplex PCR, the PCR reactions for the *CTNS* gene were replaced by a PCR reaction targeting TRECs (Figure 1). SMA screening was continued in our laboratory and the experience gathered contributed to the inclusion of SMA in the German NBS program two years later.

In October 2021, the screening for SMA officially became part of the NBS program in Germany, together with screening for sickle cell disease (SCD) (Figure 1). Therefore, we amended the multiplex *SMN1*-TREC-PCR with a PCR reaction to detect the S- and C-alleles of the β globin gene (*HBB*). With this third condition added to our multiplex panel, the commercially available assay LightMix^®^ TREC SMA HBB Newborn was established (40-0621-44, TIB Molbiol Syntheselabor GmbH, Berlin, Germany).

### 3.1. Assay Modularity

Our experience to date has demonstrated that the flexibility of multiplex PCR is especially useful for NBS, as single targets can be rapidly replaced by other targets. The pilot study setting with the combination of cystinosis [5] and SMA [4] could be quickly adapted to a combination test for SMA and SCID in the routine NBS setting without major set-up adjustments. The subsequent adaptations, including an additional test for SCD [8], could be incorporated with minimal time and effort. The modification and routine processing were completed without any complications.

### 3.2. Costs

A further advantage of multiplex assays is their cost-efficiency. The cost of employing a laboratory technician in Germany, including social security contributions and overhead costs, is approximately EUR 60,000 per year. Including the costs of consumables and enzymatic and primer–probe mixes based on their list prices, the cost per sample is estimated to be approximately EUR 2.52, provided there is an average sample throughput of around 600 samples per day (varying between 350 and 1300 samples per day). For a detailed German list price calculation, see Table 1.

Test repetitions due to technical failure or invalid results are repeated from two fresh punches on a separate plate each day, thus cost EUR 6.10 per sample repeated, which adds a total of 1.5% of costs to the annual calculation based on the numbers from the dataset presented here. There is an additional cost of the second-tier test in this case, thus HPLC for SCD is currently EUR 2.38 per sample and adds 0.5% to the annual cost calculation.

### 3.3. Cystinosis

In the first phase (pilot study), 200,901 samples were screened for cystinosis and SMA. In these samples, two patients were identified, both homozygous for the 57kb deletion in the *CTNS* gene. The modular assay allowed us to screen for both the wild-type and the mutant variants of the 57kb deletion and, furthermore, by using melting curve analysis, for the c.18_21delGACT (357DelGACT) and c.926dupG (1261InsG) mutations. Altogether, 523 samples were found to be homo- or heterozygous for at least one of the previously mentioned variants. All of these were sequenced to exclude other pathogenic variants [5]. Based on the observed allele frequencies, we estimate the sensitivity of a primary screening for a homozygous 57kb deletion in *CTNS* to be around 65%, and a screening for the heterozygous 57kb deletion plus the c.18_21delGACT and c.926dupG mutations to be 79% for the Caucasian population. After the first phase (pilot study), the *CTNS* mutation assay was replaced by the TREC assay, demonstrating the flexibility of this kind of modular assay.

### 3.4. SMA

Starting from 15 January 2018 (pilot study) to 31 May 2024, 1,001,442 samples were assessed for the c.840C>T (rs1164325688) wild-type in *SMN1*. During this period, 121 newborns with a homozygous 5q-associated SMA deletion were identified and clinically evaluated. To date, no false positive cases have been reported to us. Furthermore, neither the institutions requesting the analysis of samples from our laboratory nor the state authorities are aware of any cases missed by our test procedure. The prevalence we have seen since the introduction of SMA screening during the past 6 years is approximately 1:8300. This equates to approximately 0.012% of newborns being affected by homozygous 5q-associated SMA. An external genetic analysis of the SMA cases detected in our laboratory has identified two children with one *SMN2* copy, fifty-two children with two *SMN2* copies, thirty-one children with three *SMN2* copies, thirty children with four *SMN2* copies, five children with five *SMN2* copies and one child with six *SMN2* copies (personal communication, state authorities).

### 3.5. SCID

As of 15/08/2019, the previous multiplex combination of SMA and cystinosis was replaced by the combination of SMA and SCID assays. Therefore, one of the color channels previously used for cystinosis detection was used for TREC detection, while *SMN1* detection remained the same. From the introduction of the SCID assay in 2019 until 31 May 2024, a total of 776,754 children were screened. We identified 199 newborns requiring a control DBS card, while 15 children were classified as urgently SCID-positive. In fact, during this period, eight true SCID cases and seven severe immunodeficiencies caused by syndromes were identified and confirmed. Among the syndromic immunodeficiencies, several DiGeorge syndromes (microdeletion 22q11.2) have been diagnosed, as well as Sphingosine Phosphate Lyase Insufficiency Syndrome (SPLIS) [23], Nijmegen Breakage Syndrome [24] and Wiskott–Aldrich Syndrome (WAS) [25,26]. The 199 samples requiring a recall DBS card covered 184 newborns. The difference is due to multiple samples being taken from the same individuals, partially by other institutions. Out of the 184, 143 were diagnosed to be unaffected in the recall, 21 were deceased before a control DBS card was taken and 9 were not followed up due to reasons unknown. Two were classified as SCID-positive, where the newborns were unaffected but the mothers were under immunosuppressive treatment. A further four newborns had immunodeficiency syndromes but tested urgently positive from a first DBS, and five more had either T-cell lymphopenia or other clinical complications with recurrent infections. Up to date, no institution has informed us about an SCID case that we would have missed. Interestingly, about 60% of the cases where a control DBS was requested were premature births before a gestational age of 32 weeks. Taken together, the prevalence for SCID and syndrome-based disorders is 1:51,000 in our sample. The sensitivity of the test reached 100% for SCID and syndrome-based disorders, as up to date we do not know of any missed case. The positive predictive value was 22/195 for the samples with late amplification. This equals 11.3% for SCID or TCL. Thus, most cases were identified as unaffected in the recall. However, the urgent positive samples (no TREC amplification) were all diagnosed with SCID or a syndrome-based disorder.

### 3.6. SCD

Since the introduction of screening for SCD in October 2021, a total of 410,273 newborns were tested in our laboratory up to 31 May 2024. As a primary screening test, our test detected the genetic variants. Subsequently, all heterozygous, homozygous and irregular variants were subjected to further analysis via HPLC, which serves as a second-tier method. During this period, a total of 2122 newborns were subjected to further differentiation using the HPLC-based second-tier method. Further investigation via HPLC enabled the classification of 77 newborns as having SCD or β thalassemia. Among these newborns, 55 were identified as having a classical HbS homozygous variant, 14 exhibited a compound heterozygosity for HbS and HBC, four demonstrated a severe HbS and β thalassemia compound heterozygosity, and four exhibited other pathogenic hemoglobin variants. All of these cases were confirmed by the regional pediatric hematology center and the children received appropriate treatment before the onset of symptoms and irreversible organ damage. To date, we have not been notified by other institutions about false positive or false negative results; the prevalence is 1:5300 in our sample.

### 3.7. Turn-Around Time

Turn-around times for the molecular genetic diagnostic processes were up to 12 h from arrival to first test results in all cases; another 24 h were required if retesting or HPLC testing for SCD was required. Receiving the samples by mail requires more time. Usually, the time between sample taking and processing in the laboratory is between one and five days, two days on average.

## 4. Discussion

### 4.1. Modularity

Considering the experience accumulated over recent years in the field of molecular genetic NBS, a number of key points can be identified. The multiplexing of PCR represents an economically and ecologically valuable contribution [4,5]. Therefore, the methodology is available at low costs and easy to transfer to other countries establishing molecular genetic NBS. The simplicity of adding and exchanging individual parameters is particularly advantageous in the context of NBS. Once a routine has been implemented, it can be used for the majority of parameters without significant alterations. As described above, we have demonstrated this for the substitution of a cystinosis assay for the TREC assay, followed by the incorporation of the HBB assay.

### 4.2. Financial Outlook

The selection of diseases for NBS programs is further influenced by health economic aspects. Therefore, in addition to the individual benefits for the screened newborns, clear health economic justifications must be given. Even if Wilson and Jungner’s criteria [2] remain the gold standard for establishing novel screening programs today, their strict application would be difficult in the case of rare diseases with therapies being expensive, especially upon first approval. Keeping the cost at the time of establishment as low as possible and further reducing it as much as possible over time is a helpful concept to keep the overall costs of NBS low. The health economic burden of an SMA screening especially is significantly lower, compared to the health and social economic and environmental consequences without a screening [27]. This is also true for screening for SCID and syndrome-based disorders, where the prevalence is lower compared to SMA [28]. First, in particular, multiplexing has the potential to be expanded without any major additional costs. Second, the nucleic acid extraction technique used here allows for the inclusion of numerous other tests without additional time or sample material required.

### 4.3. Screening for SMA

As mentioned above, no false homozygous 5q-associated SMA-positive cases have been reported to us to date. Neither the specialized clinics affiliated with us nor the state authorities are aware of any case of an undetected SMA-positive child (personal communication). In our current dataset, all affected children could be reliably detected. Thus, we can report a prevalence of SMA of 1:8300 from 1,001,000 newborns tested. However, it should be noted that individual point mutations are not found in screening and will become apparent over time. In most laboratories, the detection of a heterozygous deletion has not been targeted in SMA screening. Many mutations, often de novo, can result in the absence or reduced function of the second *SMN1*-carrying allele. This is compounded by the high number of carriers in the population [29]. To screen for low-frequency variants in an estimated 5% of cases, the sequencing of the presumably intact *SMN1*-carrying allele would be necessary in all heterozygous carriers. This is not practical given the current cost pressure in screening, but might hold promising possibilities for the future. Similarly, on-site testing for SMA [30] would increase the costs and not necessarily shorten the time until treatment. Furthermore, as mentioned above, clinical manifestation is also dependent on the number of *SMN2* alleles [21,29].

### 4.4. Screening for SCID

SCID is currently one of the most complex molecular genetic screening parameters. Large numbers of genetic aberrations, including, for example, CHARGE syndrome, DiGeorge syndromes and Wiskott–Aldrich Syndrome, exhibit clinical signs and symptoms similar to those observed in SCID. This makes a clear and reliable diagnosis challenging. The common characteristic of the various SCID variants is the lack of T-cell maturation and the associated reduction in or absence of TRECs. However, the mentioned genetic disorders are likewise associated with low or absent TREC numbers [11,31]. A comparable picture has emerged in our data within the framework of studies and routine screening, particularly in relation to the frequently occurring DiGeorge syndrome. If we compare the frequency of ‘real’ SCID children with that of syndrome-based (or non-SCID) immunodeficiency, the observed frequency is almost equal in our data. However, as reflected by our data from nearly 770,000 screenings, we have identified a prevalence of 1:51,000 SCID or syndrome-based cases of immunodeficiency. This is well in accordance with previous estimates of the prevalence for SCID that predicted at least 1:100,000 cases in the United States [32] and 1:58,000 from a cohort of 3,000,000 newborns screened [32,33]. Similar estimates have been published for Japan [6]. As for SMA, the clinics affiliated with us and the state authorities are not aware of any cases of undetected SCID-positive children (personal communication). Compared to one SCID screening study from California, we had 60% less DBS recalls and 40% less urgent positive cases, while the prevalence remained comparable for SCID and syndrome-based cases [34]. The calculated positive predictive value is also in the range previously reported by other studies that range from 5 to 20% for SCID [35,36]. Better positive predictive values can only be achieved by adding other parameters to the SCID screening like kappa-deleting recombination circles (KRECs) [37]; however, the problem of low values in preterm infants remains with a combined TREC/KREC measurement [38].

### 4.5. Screening for SCD

SCD, however, is relatively easy to screen for as most mutations causing it are close together in the coding region of HBB. Nevertheless, the melting curve analysis has some advantages compared to the widely used endpoint PCR assays. The presence of other mutations causes a slight shift in the melting curve analysis, requiring the sample to be assessed using HPLC second-tier testing. We have found a prevalence of 1:5300 of SCD and SCD-related cases in 410,000 screened DBSs. This prevalence corresponds well to the prevalence for SCD as described by Tesorero [39]. However, this prevalence is way higher than estimated by previous studies [40,41,42]. A major advantage of using a melt curve analysis as described here is that rare variants can also be selected for further analyses. Thus, in our study, four pathologic HBB variants would have been missed if only the presence of one HbS allele was assessed, like for most endpoint PCR assays. In addition, the contamination risk as described in our previous study [19] is negligible for assays using melting analyses. To change a mutation-specific peak to a wild-type peak, large amounts of contaminating wild-type DNA would have to be added. Dust particles or slight DNA carryovers do not contain enough DNA to cause contamination resulting in a false or altered genotype. Taken together, this first- and second-tier testing can be recommended as it is sensitive, specific and cost-efficient, especially for high-throughput screening where HPLC cannot be applied to hundreds of samples per day in a cost-efficient manner [8].

## 5. Conclusions

Taken together, genetic NBS as presented here is highly cost-efficient due to moderate pricing and high-throughput capability. The modularity of multiplex PCR makes it easy to adapt new assays to existing laboratory equipment. Validations of novel assays or parameters can be performed quickly and either included as an ‘add-on’ to the current product version or permanently implemented in the updated product version. Unlike conventional screening methods, or even some endpoint PCR assays, it is not necessary to use different PCR mixes in separate plates to screen for these diseases. Instead, we have used multiplex techniques in a single 96-well or 384-well plate, requiring only one single DBS. This not only reduces plastic waste, but also allows us to save sampled blood for further testing or re-analysis. In addition, high-throughput pre-screening can greatly reduce the amount of the sample required for more complex, often expensive and time-consuming second-tier testing strategies, as demonstrated for the SCD assay. To date, and to our knowledge, we have achieved 100% sensitivity and specificity for homozygous *SMN1* deletion and the SCD-causing *HBB* variants and have not missed a single affected child. For SCID, to our knowledge, 100% sensitivity was observed, but the positive predictive value for SCID, TCL and syndrome-based disorders remained low around 10%.

## Figures and Tables

**Figure 1 genes-15-01467-f001:**
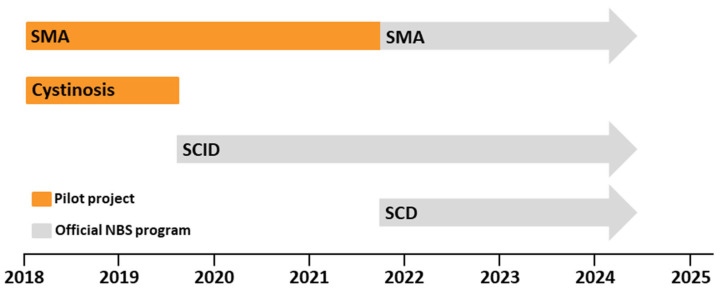
Timeline of genetic newborn screening (NBS) tests performed at Laboratory Labor Becker. Pilot project (orange) tests for spinal muscular atrophy (SMA) and cystinosis; regular tests as part of the German NBS program (gray) include, in addition, severe combined immunodeficiency (SCID) and sickle cell disease (SCD).

**Table 1 genes-15-01467-t001:** Cost of presented genetic newborn screening based on German prices.

Resources Required	Reference Number	Distributor	Price per Test (EUR)
LightMix^®^ Kit TREC SMA HBB Newborn	09 802 533 001	Roche	1.16
LightCycler^®^ Multiplex DNA Master	07 339 577 001	Roche	0.70
LightCycler^®^ 480 Multiwell Plate 384, white	04 729 749 001	Roche	0.03
Integra 12.5 µL tips, sterile	6455	Integra	0.16
Integra 1250 µL tips, sterile	6445	Integra	0.02
Other small disposables			0.05
Technician time			0.40
Total			2.52

## Data Availability

The original contributions presented in the study are included in the article, further inquiries can be directed to the corresponding authors.

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
