# Peer review of "A Modular Genetic Approach to Newborn Screening from Spinal Muscular Atrophy to Sickle Cell Disease—Results from Six Years of Genetic Newborn Screening"

_genes, 2024, doi:10.3390/genes15111467_

Round 1
Reviewer 1 Report
Comments and Suggestions for Authors
· I have a question: does the national newborn screening program only include SMA, SCID,SCD and cystinosis in Germany? Because in the intro section the authors describe: Genetic testing in the official German NBS program currently includes:…. (lines 44-50) or is a pilot study of these 3 disorders in order to include in screening program
· Line 109-110: Azenta Life Sciences, Griesheim Germany is written in other format
· Lines 199-221 and Figure 1 In 2018, we launched a large-scale pilot genetic screen for SMA and cystinosis in our 199 laboratory [4, 5, 20](Fig 1). Targets of the multiplex PCR were three genetic variants of the 200 CTNS gene for cystinosis and exon 7 in the SMN1 gene for SMA. The amplification signal 201 from one target served as an internal control for the other one. For example, a signal in 202 the PCR reaction from the CTNS gene and no amplification signal in the SMN1-PCR is a 203 valid result for an SMA patient.....etc . should be in methodology?
· How is sensitivity for detection calculated for each disease? The total number of diagnosed cases must be known.
· In clonclusion: lines 407-409: To date and to our knowledge, we have 407 achieved 100% sensitivity and specificity for homozygous SMN1 deletion, TREC and the 408 SCD-causing HBB variants and have not missed a single affected child. Where does this data come from? This data is contradictory vs. not clear with the data described in lines 251-251 where a sensitivity of 65%, 79% is described.
Author Response
We would like to thank you for your thoughtful comments and efforts to improve our manuscript.
Comment 1: I have a question: does the national newborn screening program only include SMA, SCID,SCD and cystinosis in Germany? Because in the intro section the authors describe: Genetic testing in the official German NBS program currently includes:…. (lines 44-50) or is a pilot study of these 3 disorders in order to include in screening program
Response 1: The NBS program includes SMA and SCID for genetic testing, and many more conditions for HPLC or immunoassays and other test methods. SCD can be tested using a genetic test, but can also be testing using HPLC or MALDI. Therefore we referred to SMA and SCID as part of the genetic NBS program, as they are mandatorily tested using PCR. Cystinosis is not part of the NBS program and was exclusively performed in the pilot study.
Comment 2: Line 109-110: Azenta Life Sciences, Griesheim Germany is written in other format
Response 2: Thank you for this hint, we did not see that in the original formatting and corrected it.
Comment 3: Lines 199-221 and Figure 1 In 2018, we launched a large-scale pilot genetic screen for SMA and cystinosis in our 199 laboratory [4, 5, 20](Fig 1). Targets of the multiplex PCR were three genetic variants of the 200 CTNS gene for cystinosis and exon 7 in the SMN1 gene for SMA. The amplification signal 201 from one target served as an internal control for the other one. For example, a signal in 202 the PCR reaction from the CTNS gene and no amplification signal in the SMN1-PCR is a 203 valid result for an SMA patient.....etc . should be in methodology?
Response 3: Thanks for addressing this point. As this procedure was already described in papers published before in detail, we have deleted the corresponding sentences from the manuscript.
Comment 4: How is sensitivity for detection calculated for each disease? The total number of diagnosed cases must be known.
Response 4: As mentioned in the results sections 3.4 – 3.6, no authorities reported us cases that we have missed so far for 5-q associated SMA, SCID or SCD, thus our reported number of cases equals the number of diagnosed cases. Taken together we have identified 121 cases of SMA, 15 cases of SCID and syndrome-based immunodeficiencies and 77 cases of SCD or beta-thalassemia. The total screened samples were over 1 million newborns for SMA, approximately 770,000 for SCID and over 410,000 for SCD.
Comment 5: In conclusion: lines 407-409: To date and to our knowledge, we have 407 achieved 100% sensitivity and specificity for homozygous SMN1 deletion, TREC and the 408 SCD-causing HBB variants and have not missed a single affected child. Where does this data come from? This data is contradictory vs. not clear with the data described in lines 251-251 where a sensitivity of 65%, 79% is described.
Response 5: Thank you pointing this out. To our calculations, the statements are not contradictory, as the mentioned 65% and 79% are stated regarding cystinosis and based on the numbers published before. The results for SCD, SCID and SMA are mentioned in the results sections 3.4 – 3.6 and refer to these numbers. Now we state:”To date and to our knowledge, we have achieved 100% sensitivity and specificity for homozygous SMN1 deletion and the SCD-causing HBB variants and have not missed a single affected child. For SCID, to our knowledge, 100% sensitivity was observed, but the positive predictive value for SCID, TCL and syndrome-based disorders remained low around 10%.”
Reviewer 2 Report
Comments and Suggestions for Authors
This is an excellent paper which is extremely written and presents novel data over a significant period of running the genetic NBS programme. It makes a very valuable contribution to the evidence base around genetic NBS and provides useful information for laboratories engaged in the introduction of similar programmes.
I recommend the clarification of the following minor points highlighted below.
Methods
2.4 Assay characteristics
In relation to SCID screening the authors state that “All other samples with weak amplification signals were considered for retesting”. Could they confirm how “weak amplification” was defined and how consistency in identifying repeats was maintained from batch to batch?
Results
3.2 Costs
The authors state that the costs (table 1) do not include test repetitions due to technical failure and the cost of second tier tests. I would argue that it is important to know and understand these costs. The test failure rate could have a significant impact on costs and would also be an indication of the robustness of the technology. Knowledge of the cost of second tier tests is required to understand the full cost of the screening programme – the numbers of second tier tests are included in the paper so it should be feasible to calculate the costs.
3.4 SMA
How was SMN2 copy number determined? Was testing performed on bloodspots or on a blood sample collected following referral? Presumably SMN2 testing was not performed by the screening laboratory?
Author Response
We would like to thank you for your thoughtful comments and efforts to improve our manuscript.
Comment 1:
Methods
2.4 Assay characteristics
In relation to SCID screening the authors state that “All other samples with weak amplification signals were considered for retesting”. Could they confirm how “weak amplification” was defined and how consistency in identifying repeats was maintained from batch to batch?
Response 1: Thank you for your important question. We have modified the respective sentence including more details about that topic: “All other samples with late amplification signals with Cq-values > 31 as described previously [19] were considered for retesting requesting a fresh DBS (control) card (recall) from the institution of origin.” We have further included the information on how consistency was ensured by stating: “To ensure consistency from batch to batch, the aliquot of a defined sample (positive control with plasmid DNA) was included in each run, Cq-values has to be within a range of +/- 0.5log10.”
Comment 2:
Results
3.2 Costs
The authors state that the costs (table 1) do not include test repetitions due to technical failure and the cost of second tier tests. I would argue that it is important to know and understand these costs. The test failure rate could have a significant impact on costs and would also be an indication of the robustness of the technology. Knowledge of the cost of second tier tests is required to understand the full cost of the screening programme – the numbers of second tier tests are included in the paper so it should be feasible to calculate the costs.
Response 2: Thank you for this proposal, we have included the requested calculations for the repeated qPCRs, which account for an additional overall cost of 1.5%, and 0.5% for the additional HPLC testing.
Comment 3:
3.4 SMA
How was SMN2 copy number determined? Was testing performed on bloodspots or on a blood sample collected following referral? Presumably SMN2 testing was not performed by the screening laboratory?
Response 3: You are right, we did not perform SMN2 copy number determination, as this is usually done or requested by the respective centers of treatment. Therefore we did not include any information on the details of SMN2 copy number determination, but now we have explicitly stated: “External genetic analysis of the SMA cases detected in our laboratory has identified two…”